# Global Public Health

# Awareness of management of pulmonary multidrug-resistant tuberculosis (MDR-TB) among private practitioners in suburban areas of Pune city, India: Input for developing an educational tool

Sachin Atre[1]*, Dharmendra Padalkar[1], Hanumant Chaugule[1], Trupti Sawant[1], Aryan Gupta[1], Anirvan Chatterjee[2], Maha Farhat[3,4]

1 Dr. D.Y. Patil Medical College, Hospital and Research Centre, Dr. D.Y. Patil Vidyapeeth, Pune, India, 2 Haystack Analytics Pvt Ltd, Mumbai, India, 3 Department of Biomedical Informatics, Harvard Medical School, Boston, United States of America, 4 Division of Pulmonary and Critical Care, Massachusetts General Hospital, Boston, United States of America

* atresachin2000@yahoo.com

## Abstract

Private practitioners (PPs) play a major role in caring for people with tuberculosis (TB) in India. At the same time, PPs have limited access to continuing medical education and oversight especially with regards to recent changes in the management of multi-drug resistant (MDR- TB). As a part of a larger study aimed at developing an educational tool for improving multidrug-resistant TB management, we conducted a baseline knowledge assessment of MDR-TB among PPs in suburban areas of Pune City, India. This study with a cross-sectional design was conducted during July 2022 to May 2023 among 100 PPs, who either refer and/treat TB and MDR-TB cases in Pimpri-Chinchwad Municipal Corporation (PCMC) areas of Pune in Maharashtra State. The inquiry was made using an interview schedule focused on suspicion of pulmonary TB and MDR-TB, their diagnosis and treatment. The majority of PPs were allopathic practitioners (85%) practicing in private clinics (82%). Most PPs reported that they suspect TB based on three cardinal symptoms: cough for >2 weeks (97%), fever (93%) and weight loss (82%). While 54% PPs considered the Xpert assay as the first test to diagnose MDR-TB, 32% were unaware of any test. Only 37% PPs were aware of whole genome sequencing for MDR diagnosis. A fifth of PPs selected Mantoux test use for the diagnosis of active TB. Less than a fourth of PPs knew about the second-line anti-TB drugs such as bedaquiline, delamanid or linezolid etc. and their availability either in the National TB Elimination Program (NTEP) or the private sector. Our study indicates considerable lack of awareness about pulmonary MDR-TB management among allopathic PPs in the study area and highlights the need for education and creating awareness about the same. It identified specific areas for developing an educational tool for PPs in India and elsewhere.

**Data availability statement:** All relevant data are within the paper and its Supporting Information files.

**Funding:** This work was supported through an intramural grant from Dr. D.Y. Patil Vidyapeeth [Ref. DPU/43/22 Dated 17.1.2022 to SA]. DP and HC received their salary through the study grant. The funders had no role in study design, data collection and analysis, decision to publish, or preparation of the manuscript.

**Competing interests:** The authors have declared that no competing interests exist.

## Introduction

The Global TB Report 2024 reveals following facts that are of serious concern [1]. Globally an estimated 10.8 million [95% Uncertainty Interval (UI): 10.1-11.7 million] people had TB in 2023. The net TB incidence rate between 2015 and 2023 declined with a glacial pace of 8.3%, far from the WHO End TB Strategy target of 50% reduction by 2025. While TB notification can serve as a proxy indicator for the incidence in countries with robust reporting systems as acknowledged by the WHO Global TB Report, it is not always the case particularly in high burden countries because under reporting from the private sector remains a major issue. The Global report further indicates that during COVID-19 pandemic, TB was sidelined- a reflection of which may be seen as 9% increase (from 7.5 million in 2022 to 8.2 million in 2023) in TB case notifications. Moreover, it was estimated that 400,000 (95% UI: 360000–440000) people developed multidrug-resistant or rifampicin resistant TB (MDR/RR-TB)- *i.e.,* when the bacteria have resistance to both isoniazid and rifampicin. However, only 1,75,923 (44%) were diagnosed and treated for RR/MDR-TB. The rising rates of MDR-TB is a global public health threat as one third patients do not survive and incur catastrophic costs [2]. Delays in diagnosis and subsequent initiation of effective therapy are thought to be the major drivers of MDR-TB transmission and increasing MDR-TB rates [3]. In addition, delays in care result in the advancement of TB disease severity and a higher rate of long-term complications such as chronic lung disease [4].

India carries the highest absolute (26%) global burden of TB and MDR-TB cases [1]. Although molecular diagnostics that rapidly detect TB and MDR-TB, (*e.g.,* GeneXpert TB/RIF) has been available for a decade [5], figures from the Indian Revised National TB Control Program (RNTCP) point out that these tests were used in only 50% of TB cases and often with considerable delay [6,7]. Molecular diagnostics like Xpert provides partial resistance information (to rifampicin only) while the construction of a five or more effective drug regimen for MDR-TB treatment requires a more comprehensive resistance profile. In several well-resourced settings, whole genome sequencing (WGS) is now routinely used for drug resistance diagnosis as it reduces the turn-around time and excludes the need for the conventional stepwise drug susceptibility testing (DST), which delays an accurate diagnosis. The WGS provides a resistance profile for more than twelve anti-TB drugs [8,9]. India has a thriving biotechnology sector with locally developed WHO endorsed TB diagnostics that have similar performance as the Xpert, e.g., the Truenat MTB/RIF Dx by MolBio [10]. Indian resources do exist to perform WGS with several biotechnology companies now offering these services at low cost [11].

The main barrier to optimal TB care is with the implementation and integration into the healthcare system [12]. The data are needed on the barriers that prevent the adoption and optimal use of diagnostic technologies starting with provider awareness and knowledge. Rapid referral to care and new diagnostic integration requires engagement with stakeholders in both- public and the private sector especially of front-line TB care providers to assure that every patient is rapidly referred to

appropriate care every time. The private sector in India is a large heterogeneous system of providers with limited regulation and has been documented using model 'actor' patients to pose risks for delays in care and mismanagement [13,14]. Engagement of the Indian private health sector is an established priority for improving TB patient care in India [15,16]. Leveraging our prior work on delays in care in Pune, we performed a baseline knowledge assessment of private practitioners (PPs) about TB/MDR-TB diagnosis and treatment with the aim of using findings from this assessment to design an educational tool to support improved TB care.

## Methods

### Ethics statement

The study was approved by the institutional Ethics Committee of Dr. D.Y. Patil Vidyapeeth (DYPV/EC/796/22 dated 15 Feb 2022) and Harvard Institutional Review Board (IRB) (RB22–0334 dated 10 Jan 2023). The formal written consent (except for two participants who provided an oral consent) was obtained for all the participants.

### Study design

We conducted a cross-sectional study among PPs in Pimpri-Chinchwad Municipal Corporation (PCMC), which is an industrial belt and suburban area of Pune city in the state of Maharashtra, India. We recruited and interviewed these PPs during 20th July 2022–24th May 2023. We designed and used semi-structured interviews that included 34 questions around suspecting TB/MDR-TB, diagnosis, tests used for diagnosis, pathways they adapt after diagnosis, awareness about second-line anti-TB medicines, various needs etc. for the inquiry with the PPs. We inquired also about their awareness of WGS, its benefits, costs, limitations, and views toward its implementation in Indian settings.

### Setting

Maharashtra is the second most populous state in India with the 10th highest TB incidence rate [17]. PMC and PCMC together cover the population of approximately 7.1 million and PCMC covers population of ~2 million [18]. In the year 2022, there were total 9940 (public=4282 and private=5658) cases notified from both these areas, which amounts to approximately 140 cases per 100,000 population [19]. More patients notified from private sector than public sector underscored the need for studying management of TB and/MDR-TB by the PPs in these areas. We trained two social workers who interviewed PPs after obtaining their written informed consent.

### Participants

In PCMC area, there are approximately 400 allopathic practitioners and 500 practitioners who practice alternative systems of medicine (e.g., Ayurveda, Homeopathy, Unani etc.) [20]. Since the aim of this assessment was to help design an educational tool and resources to support improved TB care, we predominantly included PPs practicing allopathic medicine who often diagnose and/treat TB and MDR-TB cases. We applied non-probability (purposive and snowball) sampling method for selecting 100 PPs that included general practitioners and chest physicians. Of 100 PPs, 98 provided a written informed consent and two provided an oral consent as they were unwilling to sign but were willing to participate in an interview. An oral consent was approved by the local institutional ethics committee for these two PPs. Following the study, we conducted a focus group discussion with 10 PPs who also participated in the study. We disseminated study findings to them, heard and documented their concerns/expectations and incorporated as input for developing an educational tool.

### Analysis

We processed the quantitative data using MS-Excel and SPSS 27.0. We calculated frequencies and percentages and presented the results as following sections: Enrolment and sample characteristics; symptoms and diagnosis of TB and

MDR-TB; awareness about MDR-TB risk and diagnostic pathway; awareness and opinions about WGS, steps that PPs take after the diagnosis of MDR-TB; counselling for patients with MDR-TB; awareness about medicines used in treating MDR-TB under the NTEP and in the private sector; and needs expressed by PPs. Further, we documented major knowledge gaps and suggested measures as input for developing an educational tool for PPs.

## Results

### Enrolment and sample characteristics

Our team approached a total 123 PPs, of which we could enrol 100 in this study. Twenty-three PPs were not enrolled for the following reasons: no time or not interested in participation (17), 'do not see TB cases' (5) and one mentioned that he has no trust in the research being conducted by the private institutes. The median age of the participants (PPs) was 43 years (IQR 36–54), and 69% were males. Of 100 PPs, the majority (85%) were allopathic practitioners with either a Batchlor of Medicine and Batchlor of Surgery (MBBS) and/or MD degree, 8% were Homeopathic practitioners and 7% were Ayurveda practitioners. Eighty two percent of PPs reported practicing in private clinics and 51% in private hospitals as well.

### Symptoms and diagnosis of TB

**Suspicion of TB.** Most practitioners reported that they suspect TB based on following symptoms: cough with duration >2 weeks (97%), fever for >2 weeks (93%) and weight loss (82%). About 20% PPs selected symptom as 'night sweats', whereas 12% selected 'blood in sputum'.

**Diagnostic tests advised by PPs for TB suspects.** For pulmonary TB, the majority (>=93%) of PPs advised chest X-ray (radiograph) and sputum microscopy; however, only 44% PPs mentioned GeneXpert (Xpert assay); 20% PPs selected Mantoux or tuberculin skin test for diagnosing active TB (Table 1). Regarding the referral place for doing tests, 31% selected government facility, 26% private facility and 43% selected either government or private facility.

PPs were also asked about the cost of diagnostic testing. Fifty seven of the 93 (61%) PPs who selected sputum microscopy as a TB diagnostic, knew that the test is free in the government sector and 48 did not know the cost in private. Others reported cost ranging from INR 100–600 (~US $1.5-7) in the private sector. Similarly, those who advise the chest X-ray (radiograph), 47 of 94 (50%) were aware of free X-ray facility in the government sector. About 11 PPs mentioned the cost of X-ray in the government centre ranged from INR 60–150 (~<2$), whereas 77 of 95 (81%) reported the cost of X-ray in the private sector ranged from INR 150–1000 (~US $2–12). Regarding the cost of Xpert assay, 20 of 44 who know about Xpert mentioned that it is free in the government/NTEP. In the private sector, the cost of Xpert assay ranged from INR 800–4500 (~<$1-$54) and 25 of 44 PPs responded that they did not know the cost in the private sector.

**Table 1. Tests advised by PPs in case of suspecting pulmonary TB.**

| Tests advised by PPs (n = 100) | % |
|---|---|
| Sputum microscopy | 93 |
| Chest X-ray (radiograph) | 94 |
| Gene Xpert/CBNAAT | 44 |
| CT Chest | 26 |
| Mantoux test | 20 |
| True NAAT and bronchoscopy | 8 |

*Note: Multiple responses*

## Awareness about MDR-TB risk and diagnostic pathway

Sixty-one PPs reported that they provide care (diagnose, refer and/treat) for MDR-TB cases. When asked about risk factors for suspecting MDR-TB, less than half of PPs selected each of prior history of MDR and non-response to treatment; however, only 6 PPs reported that they consider both these risk factors. Only a quarter of PPs suspected MDR after an incomplete treatment course. However, only 8 PPs each selected that they also combine incomplete treatment course with either prior history of TB or non-response to the treatment. Only 5% selected contacts with MDR-TB cases as a criterion for suspecting MDR-TB (Table 2). When asked about the first test to diagnose MDR-TB, 54% PPs selected the Xpert assay, 11% selected culture-based drug susceptibility and 32% selected 'do not know' (Table 3).

## Awareness and opinions about the WGS

Of the 100 PPs, only 37 were aware of WGS for resistance diagnosis. Initially 16 of 37 reported that they refer patients for WGS. However, when asked about the turnaround time required for getting the WGS test results, 6 reported >14 days, whereas 7 reported between 7–14 days. When inquired further, the research team found that remaining 3 PPs erroneously referred 'Xpert assay' as 'WGS' as they reported receipt of report within <6 days (which is generally Xpert assay report). Thus, 13 of 37 PPs reported that they refer patients for the WGS. Furthermore, only 37 PPs who had some awareness, indicated interest in learning more about the WGS. Of these, 11 PPs expressed concerns about the cost, 7 about the complexity and 4 about the accuracy of WGS. Overall, 63 PPs did not provide any opinion/indicate interest in learning/knowing more about the WGS.

## Steps that PPs take after the diagnosis of MDR-TB

As far as cases of MDR-TB are concerned, 61 of 100 PPs reported that they deal with them. Remaining 39 PPs did not see MDR-TB cases but they directly refer them either to a chest physician or the NTEP. Twenty nine of 61 PPs reported that they refer the case to other private practitioners once MDR-TB diagnosis happens; 18 reported that they prescribe treatment at their own clinics and 14 reported that they refer the case to the NTEP. When inquired about reasons for

**Table 2. Criteria for suspecting MDR-TB.**

| Criteria (n = 100) | % |
| --- | --- |
| Prior history of TB/MDR-TB | 43 |
| Non-response to treatment (first line) | 42 |
| Incomplete treatment course | 24 |
| Laboratory reports | 14 |
| Physical condition of patient | 12 |
| Contact with MDR-TB case | 5 |

*Note: Multiple responses*

**Table 3. First test used to diagnose MDR-TB.**

| Tests (n = 100) | % |
| --- | --- |
| Xpert assay/CBNAAT | 54 |
| Conventional Drug Susceptibility Testing (DST) | 11 |
| Line probe | 2 |
| CT chest | 1 |
| Don't know | 32 |

referral to other PPs, expert opinion was the main reason (22 of 29), followed by lack of capacity to manage (6) and/ patient's decision (6). Interestingly, 22 of 29 who reported referrals to other PPs, all of them reported that patients often come back to them mainly for two reasons: being their family physician (18 of 22) and/ when they experience side-effects of treatment (6 of 22). Similarly, 35 PPs who refer MDR-TB cases to the NTEP also reported that patients do return to them for the same reasons.

### Counselling for patients with MDR-TB

Overall, 76 PPs reported that they counsel patients who have MDR-TB irrespective of whether they treat them or not. Following responses were received when asked about areas on which counselling is done: the importance of treatment compliance (100%); diet (60%); side-effect management (34%) and the psychosocial support (28%).

### Awareness of medicines used for treating MDR-TB

When asked about medicines used for treating MDR-TB, less than a fifth of PPs were aware of their availability in the NTEP and less than a fourth knew about the availability in the retail pharmacies. Awareness about medicines such as Bedaquiline and Linezolid that constitute BPaL regimen was extremely low (Table 4).

### Needs expressed by PPs

When asked about the needs that PPs felt, 76% mentioned that they want information on updated guidelines for TB and MDR-TB; 48% mentioned education on how to diagnose MDR-TB; 19% mentioned strengthening reporting and referral system with NTEP; 16% expressed the need for increasing patient awareness and counselling and 12% mentioned about increasing awareness of WGS. During the focus group discussion, PPs expressed that they are willing to take out time to get training on MDR-TB diagnosis and management. They are also willing to coordinate with the NTEP provided they receive proper feedback about the patients they refer there.

## Discussion

Despite known for weak regulation, the private sector undoubtedly continues to have a major role in TB management in India for past several decades [17,21,22]. In this study of 100 PPs who provide TB care in an urban and industrial settlement of PCMC, we found significant knowledge gaps regarding the suspicion, diagnosis and treatment of pulmonary TB

**Table 4. PP's awareness about availability of second-line medicines for treating MDR-TB.**

| Availability of medicines for MDR-TB (n = 100) | In the NTEP (%) | In retail pharmacies (%) |
|---|---|---|
| Bedaquiline | 19 | 13 |
| Delamanid | 12 | 6 |
| Pretomanid | 0 | 2 |
| Linezolid | 12 | 21 |
| Levofloxacin | 21 | 25 |
| Moxifloxacin | 16 | 22 |
| Clofazimine | 12 | 13 |
| Aminoglycosides | 9 | 10 |
| Capreomycin | 14 | 13 |
| Ethionamide | 15 | 16 |
| Prothionamide | 4 | 4 |
| Cycloserine | 15 | 19 |
| Terizidone | 1 | 3 |

and MDR-TB. Although we identified high awareness of the three TB cardinal symptoms -cough, fever and weight loss, other important symptoms such as hymoptysis and night sweats were not selected as suspicious of pulmonary TB, indicating a less nuanced awareness of TB's manifestations.

Sputum microscopy and chest-X-ray remain the most selected TB diagnostic approaches by the PPs. This observation has several important implications. Sputum microscopy has high specificity (98%) but low sensitivity (20–60%) which is likely to result in missing individuals who may have TB [23]. On the contrary, chest X-rays have high sensitivity (96%, IQR 93–98%) but moderate specificity (46% IQR 35–50%) [24]. Such moderate specificity can lead to over diagnosis of pulmonary TB and result in an unnecessary treatment. Surprisingly, some PPs were not able to tell the costs of microscopy and X-rays in the private laboratories, where they frequently refer their patients. It is hard to believe that they are unaware of the costs. More in-depth interaction with the PPs may be needed to explore the reasons behind such nondisclosure. Also, a considerable number of PPs were unaware about free availability of these basic tests in the NTEP/government centres.

The standard of diagnosis for MDR-TB involves clinical suspicion and rapid molecular testing for rifampicin resistance using Xpert assay. Despite the recommendation of Xpert/CBNAAT by the WHO and Indian TB guidelines as a first-line diagnostic for TB suspects and having current wide availability (approximately >10 facilities) since 2018 onward in the PCMC area, less than half of PPs selected Xpert/CBNAAT as the first test for TB suspect. It may suggest that many PPs are still under a misconception that Xpert assay is meant only for diagnosing MDR-TB and not for the drug susceptible TB, which needs to be clarified. Moreover, only 54% identified it as the first test for diagnosis of MDR-TB, which points out lack of knowledge of this test among the remaining practitioners. In such situations, patients incur unnecessary out of pocket expenditure for diagnosis until they reach the NTEP. Our recent study among patients with MDR-TB in Pune city documented significant out of pocket expenditure that incurred in the private sector (which was reported as the first source of help seeking by 68% patients) prior to receiving MDR-TB diagnosis in the NTEP and a few had incurred catastrophic costs [17]. A study by Mullerpattan et al. in Mumbai reported that treatment cost for pulmonary MDR-TB in the private sector was exorbitant with an average $5723 [25]. The concern had been raised about price variations for Xpert assay across different countries. Puri et al. reported about Initiative for Promoting Affordable and Quality TB Tests (IPAQT)- a private sector initiative, which offers WHO-approved diagnostics at concessional fixed price of INR 2000 ($30·26), compared with an average of $52·82 in the rest of the private sector in India [26]. Furthermore, the lack of awareness about Xpert assay can result in delayed diagnosis of TB and MDR-TB which eventually can lead to transmission of resistant TB strains in the communities. Our previous study supports this assertion. It reported the median diagnostic delay (onset of symptoms to diagnosis) for patients with MDR-TB as 90 days (IQR 60–180) and 68% (87 of 128) of them reported private sector as the first point of help seeking. Similarly, even among non MDR patients the median diagnostic delay was 60 days (IQR 30–90) and for 100 of 139 patients, private sector was the first point of help seeking [17]. In case of government sector/the NTEP, the diagnosis of TB and RR/MDR-TB generally happens within 3 days, and things have improved especially after the rollout of Xpert assay.

With reference to the awareness of the latest diagnostic method- WGS, two-thirds of PPs were unaware of it and its role in expediting the diagnosis of drug-resistant TB. Also, only a few PPs expressed interest in learning more about it. Given such situation, it is likely that the diagnosis of severe forms of drug-resistant TB such as pre-extensively drug-resistant TB (pre-XDR-TB) and XDR-TB can be easily missed or delayed. It will not only deteriorate patient's health but can lead to further amplification of resistance and transmission of highly drug-resistant TB strains in the community causing threat to public health. Regarding the availability of second-line medicines, we found that only a fourth of the PPs could name them and majority were unaware of the availability either in the NTEP or retail pharmacies. Such a situation among allopathic PPs in a well-connected urban setting like Pune city raises concerns about the quality of TB/MDR-TB management in distantly located rural and/tribal areas of India with less connectivity and scarce resources.

As the private sector is the first point of contact for majority of patients with TB in India often resulting in care delays compared with the public sector, assessing PPs' awareness and practices related to TB and MDR-TB is an important step.

Previous studies by Udwadia et al. [22] and Yadav et al. [27] documented poor awareness and knowledge of MDR-TB among PPs. Despite technological innovations, and the government now mandating and incentivizing reporting of TB by PPs, the present study indicates that awareness of best practices for suspecting and diagnosing TB and/MDR-TB has not improved. However, besides identifying exact areas for developing an educational tool for PPs, our study has identified areas where PPs can work with the NTEP. Recently, the WHO has endorsed targeted next-generation sequencing (tNGS) for the diagnosis of drug-resistant tuberculosis [28]. During the focus group discussion with PPs, we showed them the WGS sample reports that simply mention resistance/sensitivity to a particular drug and it appealed to them. This method may be useful for introducing tNGS and WGS to PPs to generate interest without getting into the technical complexities. Further, many PPs mentioned that patients often approach them for seeking advice on food and side-effects as they lack guidance at the NTEP. In our opinion, efforts should be made by the NTEP to involve PPs in their activities so that patients can be offered better care. Finally, our study provides input for developing an educational tool, which we hope will be useful for other developing countries as well, where the private sector plays an important role in TB care (Table 5). As a next step, based on this experience, we plan to develop and pilot test our educational tool for PPs.

## Limitations of the study

While our study offers insights to awareness of PPs regarding pulmonary TB/MDR-TB management, it has several limitations. First, we primarily sampled approximately 21% (85 of 400) of allopathic PPs from the study area who actively manage and/or notify TB cases to the NTEP. However, the number of PPs from alternative systems in the study area as reported in a previous study is large [20]. Due to limited funding and short duration of the study, we had to adapt non-probability sampling method. Thus, our study is prone to selection bias to some extent. Second, there are numerous PPs belonged to alternative systems such as Ayurveda and Homeopathy but manage TB cases through allopathic practice. Given the limited resources, we could include only a few known PPs from them. Thus, we could get limited insights of that group of PPs. Third, we did the study among PPs in urban areas. Thus, the generalisability of the findings may be limited. Nevertheless, studying awareness among PPs and their practices related to TB/MDR-TB care in rural areas would be important, since their access to information and technology is limited. We strongly believe that the identified common gaps and suggested measures will help in contributing to the development of an educational tool that will have wider applicability.

**Table 5. Gap analysis and input for developing an educational tool for PPs.**

| Identified knowledge gaps | Inputs for the educational tool for PPs along with the NTEP support |
|---|---|
| 1. Lack of knowledge about TB symptoms other than cough, fever, weight loss | Education on TB symptoms other than the cardinal ones (e.g., night sweats, hymoptysis etc.) |
| 2. One fifth of the PPs reported that they rely on Mantoux test for TB diagnosis | Education on discouraging the use of Mantoux test for the diagnosis of active TB |
| 3. Lack of knowledge on suspecting MDR-TB resulting in delayed diagnosis and treatment | Education on using laboratory reports, suspecting MDR-TB and inquiry about contacts of MDR-TB cases to prevent diagnostic delays |
| 4. Lack of knowledge about basic tests (e.g., Xpert assay) used in MDR-TB diagnosis and costs of tests. | Education on the Xpert assay as the point of care diagnostics for detecting TB and RR/MDR-TB and cost of the tests |
| 5. Lack of knowledge about WGS or genome sequencing, no motivation to learn more about it as it is not introduced to them. | Sessions to introduce WGS or tNGS and its importance in early diagnosis of TB and DR-TB for initiating timely and correct treatment to prevent the transmission of resistant strains in the community; motivational steps (incentives) to encourage referral for tNGS or WGS for timely diagnosis of drug resistant-TB. |
| 6. Patients often return from NTEP to PPs for the advice on psychosocial issues, food, side-effect management etc. | Training for PPs on handling psychosocial issues, nutrition and side effects of the treatment. |
| 7. Lack of knowledge about novel anti-TB drugs such as Bedaquiline, Pretomanid and Linezolid (BPaL regimen) or other second-line drugs | Education on availability of second-line anti-TB drugs both at NTEP and private sector, training on updated guidelines on MDR-TB management, update on BPaL regimen etc. |

## Supporting information

**S1 Data. S1_data.sav.**
(SAV)

## Acknowledgments

We gratefully acknowledge the funding support from Dr. D.Y. Patil Vidyapeeth/University (DPU) Pune. We thank all the study participants for their support. We are grateful to Dr. Bhawalkar, Prof. Hetal Rathod and Prof. Abhijit Tilak for their support in smooth execution of the study. We thank Dr. Barthwal for the administrative support in the initial one month of the study.

## Author contributions

**Conceptualization:** Sachin Atre, Hanumant Chaugule, Anirvan Chatterjee, Maha Farhat.

**Data curation:** Sachin Atre, Dharmendra Padalkar, Hanumant Chaugule, Trupti Sawant, Aryan Gupta, Anirvan Chatterjee.

**Formal analysis:** Sachin Atre, Dharmendra Padalkar, Trupti Sawant, Aryan Gupta, Maha Farhat.

**Funding acquisition:** Sachin Atre.

**Methodology:** Sachin Atre, Dharmendra Padalkar, Hanumant Chaugule, Trupti Sawant, Aryan Gupta.

**Supervision:** Sachin Atre, Maha Farhat.

**Writing – original draft:** Sachin Atre.

**Writing – review & editing:** Sachin Atre, Dharmendra Padalkar, Hanumant Chaugule, Trupti Sawant, Aryan Gupta, Anirvan Chatterjee, Maha Farhat.

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
