## [Decision Letter · Decision Letter 0]

24 Mar 2025

PGPH-D-24-02970

Awareness of management of pulmonary multidrug-resistant tuberculosis (MDR-TB) among private practitioners in Pune city, India: Input for developing an educational tool

Dear Dr. Atre,

Thank you for submitting your manuscript to PLOS Global Public Health. After careful consideration, we feel that it has merit but does not fully meet PLOS Global Public Health’s publication criteria as it currently stands. Therefore, we invite you to submit a revised version of the manuscript that addresses the points raised during the review process.

We look forward to receiving your revised manuscript.

Kind regards,

Raquel Muñiz-Salazar, Ph.D.

Academic Editor

Journal Requirements:

i. Please clarify all sources of funding (financial or material support) for your study. List the grants (with grant number) or organizations (with url) that supported your study, including funding received from your institution. 

ii. State the initials, alongside each funding source, of each author to receive each grant.

iii. State what role the funders took in the study. If the funders had no role in your study, please state: “The funders had no role in study design, data collection and analysis, decision to publish, or preparation of the manuscript.”

iv. If any authors received a salary from any of your funders, please state which authors and which funders.

3. Please make sure the funding information on the submission form matches your financial disclosure statement. Please indicate by return the full and correct funding information for your study and confirm the order in which funding contributions should appear. Please be sure to indicate whether the funders played any role in the study design, data collection and analysis, decision to publish, or preparation of the manuscript.

4. Please insert an Ethics Statement at the beginning of your Methods section, under a subheading 'Ethics Statement'. It must include:

1) The name(s) of the Institutional Review Board(s) or Ethics Committee(s)

2) The approval number(s), or a statement that approval was granted by the named board(s) 

3) (for human participants/donors) - A statement that formal consent was obtained (must state whether verbal/written) OR the reason consent was not obtained (e.g. anonymity). NOTE: If child participants, the statement must declare that formal consent was obtained from the parent/guardian.

5. In the online submission form, you indicated that “Data are available with the first author. Required data are presented in the form of tables.”. 

3. Uploaded as supplementary information.

Additional Editor Comments (if provided):

Both reviewers have recommended the acceptance of the manuscript; however, one reviewer advocates for minor revisions, while the other suggests major revisions.

It is crucial to address all recommendations from the reviewers to meet the journal's performance standards.

Reviewers' comments:

Reviewer's Responses to Questions

**Comments to the Author**

1. Does this manuscript meet PLOS Global Public Health’s publication criteria ? Is the manuscript technically sound, and do the data support the conclusions? The manuscript must describe methodologically and ethically rigorous research with conclusions that are appropriately drawn based on the data presented.

Reviewer #1: Yes

Reviewer #2: Yes

2. Has the statistical analysis been performed appropriately and rigorously?

Reviewer #1: Yes

Reviewer #2: Yes

3. Have the authors made all data underlying the findings in their manuscript fully available (please refer to the Data Availability Statement at the start of the manuscript PDF file)?

Reviewer #1: Yes

Reviewer #2: No

4. Is the manuscript presented in an intelligible fashion and written in standard English?

Reviewer #1: Yes

Reviewer #2: Yes

5. Review Comments to the Author

Reviewer #1: Among the key words include India

Line 60: The TB case notifications increased by 9% (from 7.5 million in 2022 to 8.2 million in 2023) making TB the again the lead infectious killer surpassing COVID-19. Consider rewriting this sentence. It is not coherent.

Line 78: WGS resistance diagnosis as it cuts costs and turn-around time. I would think that this may not be entirely true. Please rewrite this sentence.

Line 139: Our team approached a total 123 PPs, of which we could enrol 100 in this study. What is the estimated total number of PP in Pune city, India so we can know the proportion represented by 100 PPs. I would have liked to know the number of suspected TB patients they see on average daily or monthly.

Line 178: further, the research team found that some PPs were erroneously referring ‘Xpert assay’ as ‘WGS’. What percentage of PPs was this?

Discussion

I am interested in comparing the turnaround time for TB diagnosis between private and government facilities. Please if you can provide details on this.

Also comment on the standard of diagnosis for MDR-TB in India

Comment about the cost of treating MDR TB in PP versus government.

Reviewer #2: The manuscript is well-written and presents an important issue regarding the role of private practitioners in TB care in India. The research topic is highly relevant to public health, especially considering the significant burden of TB and MDR-TB in the country.

However, I have two main concerns that need to be addressed:

First, the sample size of 100 private practitioners seems relatively small, which may limit the generalizability of the findings. In the methodology section, you should provide a more detailed description of the sampling process. How were the participants selected? What measures were taken to ensure the representativeness of the sample? A clearer explanation of the sampling strategy would strengthen the study's credibility.

Second, there is a lack of discussion on study limitations: The discussion section does not adequately acknowledge the study's limitations. It is important to acknowledge potential limitations such as the small sample size, potential selection bias, and any other factors that may have influenced the results. Discussing these limitations will provide a more balanced interpretation of the findings and demonstrate a deeper understanding of the research's strengths and weaknesses.

6. PLOS authors have the option to publish the peer review history of their article (what does this mean? ). If published, this will include your full peer review and any attached files.

**Do you want your identity to be public for this peer review?** For information about this choice, including consent withdrawal, please see our Privacy Policy .

Reviewer #1: No

Reviewer #2: No

---

## [Decision Letter · Decision Letter 1]

31 Aug 2025

Awareness of management of pulmonary multidrug-resistant tuberculosis (MDR-TB) among private practitioners in suburban areas of Pune city, India: Input for developing an educational tool

PGPH-D-24-02970R1

Dear Dr. Atre,

We are pleased to inform you that your manuscript 'Awareness of management of pulmonary multidrug-resistant tuberculosis (MDR-TB) among private practitioners in suburban areas of Pune city, India: Input for developing an educational tool' has been provisionally accepted for publication in PLOS Global Public Health.

Best regards,

Raquel Muñiz-Salazar, Ph.D.

Academic Editor

The manuscript shows significant improvement in clarity, methodology, and the presentation of results.

I am pleased to inform you that your manuscript has been accepted for publication, pending minor editorial adjustments (e.g., formatting, final reference checks, or minor language edits).

Reviewer Comments (if any, and for reference):

Reviewer's Responses to Questions

**Comments to the Author**

1. If the authors have adequately addressed your comments raised in a previous round of review and you feel that this manuscript is now acceptable for publication, you may indicate that here to bypass the “Comments to the Author” section, enter your conflict of interest statement in the “Confidential to Editor” section, and submit your "Accept" recommendation.

Reviewer #1: All comments have been addressed

2. Does this manuscript meet PLOS Global Public Health’s publication criteria ? Is the manuscript technically sound, and do the data support the conclusions? The manuscript must describe methodologically and ethically rigorous research with conclusions that are appropriately drawn based on the data presented.

Reviewer #1: Yes

3. Has the statistical analysis been performed appropriately and rigorously?

Reviewer #1: Yes

4. Have the authors made all data underlying the findings in their manuscript fully available (please refer to the Data Availability Statement at the start of the manuscript PDF file)?

Reviewer #1: Yes

5. Is the manuscript presented in an intelligible fashion and written in standard English?

Reviewer #1: Yes

6. Review Comments to the Author

Reviewer #1: The authors have adequately addressed all my previous comments and concerns. The revised manuscript demonstrates substantial improvement in clarity, methodology, and presentation of results. I have no further suggestions or outstanding issues.

7. PLOS authors have the option to publish the peer review history of their article (what does this mean? ). If published, this will include your full peer review and any attached files.

**Do you want your identity to be public for this peer review?** For information about this choice, including consent withdrawal, please see our Privacy Policy .

Reviewer #1: **Yes: ** Gerald Mboowa
